Synergistic effects of selenium nanoparticles and LED light on enhancement of secondary metabolites in sandalwood (Santalum album) plants through in-vitro callus culturing technique

Mazhar Muhammad Waqas 1
Ishtiaq Muhammad drishtiaq.bot@must.edu.pk 1 2
Maqbool Mehwish 1 2
Jafri Faisal Iqbal 3
Siddiqui Manzer H. mhsiddiqui@ksu.edu.sa 4
Alamri Saud 4
Akhtar Mohd Sayeed 5
1 Mirpur University of Science and Technology, Mirpur Pakistan , Mirpur , Pakistan
2 Department of Botany, Climate Change Research Centre, Herbarium and Biodiversity Conservation Labortary, Azad Jammu and Kashmir University of Bhimber (AJKUoB) , Bhimber , Pakistan
3 University of Gujrat , Gujrat , Pakistan
4 Department of Botany and Microbiology, College of Science, King Saud University , Riyadh , Saudi Arabia
5 Department of Botany, Gandhi Faiz-E-Aam College , Shahjahanpur , Uttar Pradesh , India
Domingues Douglas
Electronic publication date: 2024 Sep 26
Publication date: 2024
Volume: 12
Electronic Location ID: e18106
Received 2024 Apr 10; Accepted 2024 Aug 27
Copyright: ©2024 Mazhar et al.
Copyright year: 2024
Copyright holder: Mazhar et al.
License: This is an open access article distributed under the terms of the Creative Commons Attribution License, which permits unrestricted use, distribution, reproduction and adaptation in any medium and for any purpose provided that it is properly attributed. For attribution, the original author(s), title, publication source (PeerJ) and either DOI or URL of the article must be cited.
License URL: https://creativecommons.org/licenses/by/4.0/

Keywords: Elicitation, Tissue culture, Bioactive antioxidants, Seleniumnanoparticle, Sandalwood

Funding: Researchers Support Project RSP2024R194 This work was supported by the Researchers Supporting Project number (RSP2024R194), King Saud University, Riyadh, Saudi Arabia. The funders had no role in study design, data collection and analysis, decision to publish, or preparation of the manuscript.

==============================
The yield and concentration of secondary metabolites (SMs) in plants can vary due to numerous challenges such as dynamic environmental conditions, moisture, soil quality, soil organic matter and plant genetics. To obtain a good yield of SMs novel elicitation approaches, such as the use of biotic and abiotic stressors, genetic modifications, and optimized growth conditions, have been practiced, particularly the use of selenium nanoparticles (SeNPs) and light emitting diode (LED) interaction through employing tissue culture technique. In the present study, in vitro callus cultures of sandalwood (Santalum album L.) were subjected to elicitation with different concentrations of SeNPs with doses of 30 µg/L, 60 µg/L, and 90 µg/L in combination with green (∼550 nm), red (∼660 nm) and blue (∼460 nm) LED lights. Interaction of these treatments produced 16 treatments replicated three times in 48 test tubes. The results were analysed using two-way ANOVA and Tukey’s HSD test. The study revealed that synergistic interaction between SeNPs and LED light wavelengths significantly enhanced callus growth and secondary metabolite (SM) production eliciting callus cultures with blue LED light and a dose of 90 µg/L SeNPs resulted in an increase in callus growth including fresh weight, dry weight, and the number of shoot branches per callus. This combined treatment positively influenced the functions of major bioactive antioxidants such as superoxide dismutase (SOD), peroxidase (POD), catalase (CAT) and phenylalanine ammonia-lyase (PAL). Furthermore, the concentrations of essential secondary metabolites, including total phenolic, total saponins, casein/BSA/PVPP-bound tannins, flavan-3-ols, and tocopherols experienced substantial elevation under the synergistic influence of SeNPs and LED light conditions. The sandalwood plants produced through the callus culturing technique using optimized SeNPs and LED lights show an enhanced yield of secondary metabolites, which will be very useful and potential for pharmaceutical, cosmetic and various other industries to discover and develop novel products.

Introduction

Sandalwood (Santalum album L.) is a fragrant tropical tree native to the Indian subcontinent, Australia, and Southeast Asia (Teixeirada Silva et al., 2016). It is renowned for its highly aromatic heartwood, which is used to extract sandalwood oil. This oil has been used for centuries in perfumes, cosmetics, and traditional medicines due to its pleasant aroma and therapeutic properties (Swamy, 2021). Sandalwood is also valued in religious and spiritual practices in various cultures. The tree is slow-growing and its heartwood is rich in essential oils, making it a precious and sought-after resource (McLellan, Dixon & Watson, 2021).

Sandalwood contains a variety of important secondary metabolites, particularly in its heartwood (Kumar et al., 2019). These compounds contribute to the oil’s medicinal value. Sandalwood oil has been used in traditional medicine for its various health benefits. It is believed to have antimicrobial, anti-inflammatory, and calming effects, making it popular in aromatherapy and natural remedies (Bisht et al., 2020).

In vitro, callus cultures of sandalwood have been explored as a promising approach for enhancing the production of secondary metabolites, particularly sandalwood oil and its related compounds (Misra & Dey, 2012). Callus cultures are undifferentiated cells that are grown in a controlled environment, providing researchers with a system to manipulate and study plant biochemistry without harming the whole plant (Sardar et al., 2023). Callus cultures can be supplemented with precursor molecules that are essential for the biosynthesis of sandalwood oil compounds. By providing these precursors in higher concentrations, the production of target metabolites can be enhanced (Mushtaq et al., 2023).

Elicitation techniques can be employed to boost the synthesis of sandalwood oil components (Maqbool et al., 2023). Nanoparticles, especially selenium nanoparticles (SeNPs), have gained significant attention in plant science as elicitors–substances that induce stress responses in plants, leading to enhanced production of secondary metabolites (Rivero-Montejo, Vargas-Hernandez & Torres-Pacheco, 2021). Selenium nanoparticles, due to their small size and high surface area, can efficiently penetrate plant cells and activate various defense mechanisms (Joshi, De Britto & Jogaiah, 2021). SeNPs used as elicitors have been shown to stimulate the synthesis of secondary metabolites, including antioxidants and phytochemicals, in plants. This enhanced production is a result of the plant’s response to the stress caused by the nanoparticles, leading to increased bioactive compound levels. Additionally, selenium is an essential micronutrient for plants (Ishtiaq et al., 2023), making SeNPs a dual-purpose elicitor by enhancing both nutrient uptake and secondary metabolite biosynthesis. The use of light-emitting diode (LED) lights in plant cultivation has revolutionized the way researchers and growers optimize the production of phytochemicals. LED lights are energy-efficient and highly controllable, allowing for precise manipulation of light spectra, intensity, and duration (Taulavuori et al., 2017). By tailoring these factors to specific wavelengths, plants can be stimulated to produce higher quantities of desirable phytochemicals. For example, certain wavelengths of light, such as ultraviolet (UV) and blue light, have been found to enhance the biosynthesis of flavonoids, anthocyanins, and antioxidants, while red light promotes the production of compounds like carotenoids and chlorophyll (Manivannan et al., 2015).

The current landscape of sandalwood cultivation faces formidable challenges that necessitate innovative solutions. Conventional cultivation approaches often struggle to optimize the production of valuable secondary metabolites in sandalwood. The reliance on traditional methods without addressing the intricate environmental and genetic factors limits the reliability and efficiency of sandalwood cultivation. This gap becomes particularly pronounced in the context of meeting the escalating demand for bioactive compounds from sandalwood, which are crucial for various industrial applications (Taulavuori et al., 2017). Additionally, the limited understanding of elicitation techniques in the context of sandalwood further accentuates the challenges. The absence of tailored strategies that harness the synergistic effects of SeNPs and LED light on sandalwood callus cultures reflects a significant gap in the current knowledge base. Addressing these challenges is imperative for the sustainable and efficient cultivation of sandalwood. The innovative approach presented in this study fills a critical void by introducing a novel elicitation technique that not only enhances metabolite production but also provides insights into optimizing the growth parameters of sandalwood callus cultures. By bridging these gaps, the study lays the foundation for a more robust and reliable sandalwood cultivation methodology, thereby offering a transformative solution to the challenges currently faced in the field (Cioć et al., 2018).

The primary objective of this study is to investigate the hypothesis that the combined application of SeNPs and LED light can significantly enhance the production of secondary metabolites (Gupta & Sood, 2023) in vitro cultures of sandalwood. To achieve this, the study is designed to evaluate both the individual and synergistic effects of SeNPs and LED light on the growth parameters and secondary metabolite production of sandalwood callus cultures. The research specifically aims to optimize the concentration of SeNPs by determining the most effective elicitation dose through a systematic exploration of varying concentrations. The study further aims to quantify and characterize key secondary metabolites to provide a detailed understanding of the biochemical changes induced by the combined elicitation approach. Ultimately, by addressing these objectives, the research endeavors to contribute valuable insights to the field of plant biotechnology, offering practical advancements for optimizing sandalwood cultivation practices and enhancing the overall efficiency of secondary metabolite production in vitro cultures.

Material and Methods

Plant material collection

Fresh shoots of Sandalwood plant weighing 0.5 g were aseptically placed onto Murashige and Skoog (MS) medium in glass jars. The medium was supplemented with indole-3-butyric acid (IBA) at a concentration of 3 mg L−1 and kinetin (KN) at a concentration of 1 mg L−1. Additionally, the medium contained 0.8% agar-agar and 3% w/v sucrose. The pH of the medium was adjusted to 5.6 before autoclaving. The cultures were incubated under controlled conditions at a temperature of 18 ± 1 °C and a relative humidity of approximately 76%. The photoperiod was set to 16 h of daylight and 8 h of darkness. Sandalwood plants are adapted to moderate temperatures, and the chosen range of 18 ± 1 °C is close to the ideal conditions found in their native habitats. The 16-hour daylight period provides sufficient light for photosynthesis, which is crucial for growth and biomass accumulation. The 8-hour dark period allows for essential rest and recovery processes, promoting overall plant health and development. The entire procedure was carried out under aseptic conditions to prevent contamination following Sardar et al. (2023).

Elicitation experiment and treatment layout

A variety of colored 12-watt LED lamps were employed, purchased from Amazon (Seattle, WA, USA), including green, red, and blue lights (Empire brand, Tejas brand), with white fluorescent light serving as the control. Sandalwood plants were randomly assigned to each LED treatment within the plant tissue culture lab, maintained at a temperature of 15 °C ± 1 °C with a photoperiod of 16 h of light and 8 h of darkness (Gupta & Sood, 2023).

Dark orange to dark red SeNPs obtained from Sigma Aldrich (https://www.sigmaaldrich.com/specification-sheets/494/341/919519-BULK.pdf; Product No. 919519). According to suppliers, the SeNPs were characterized using inductively coupled plasma–mass spectrometry (ICP–MS) to confirm the elemental composition, verifying the presence of selenium, having size distribution of SeNPs, ranging between 70 and 90 nm. Different SeNP concentrations, including 30, 60, and 90 µg L−1 was prepared for elicitation treatments, with a 0 µg L−1 concentration serves as the control.

The SeNPs were dispersed in phosphate-buffered saline solution to enhance their solubility and stability. This buffer was chosen for its ability to maintain the pH and ionic strength, which are critical for stabilizing SeNPs and preventing aggregation. The stock solution of SeNPs was prepared at a concentration of 1,000 µg/L. The nanoparticles were initially dispersed in the buffer solution using ultrasonication. This process involved subjecting the suspension to ultrasonic waves for 30 min to break down any agglomerates and ensure a homogeneous suspension. For the preparation of different SeNP concentrations (30, 60, and 90 µg/L), precise volumes of the stock solution were transferred to separate containers. The following volumes of the stock solution were used:

• 30 µg/L: (3 mL of the stock solution diluted to 100 mL with the buffer)

• 60 µg/L: (6 mL of the stock solution diluted to 100 mL with the buffer)

• 90 µg/L: (9 mL of the stock solution diluted to 100 mL with the buffer)

Each diluted solution was thoroughly mixed using a vortex mixer to ensure even dispersion of the nanoparticles. This step was crucial to maintain the consistency of the SeNP concentration in each sample. The specific volumes of the prepared SeNP solutions were applied to the experimental setups as per the standardized protocol described in the methods section (Sardar et al., 2023).

Table 1 Experimental sketch of the study depicting treatment plans and interactions.

Treatments a	Treatments in column	
		Control	Green LED	Red LED	Blue LED	
Treatment in rows	0 µg/L	1	5	9	13	
	30 µg/L	2	6	10	14	
	60 µg/L	3	7	11	15	
	90 µg/L	4	8	12	16	
Notes.

a The table illustrates the treatment plans utilized in the experiment, showcasing the interaction of four treatments arranged in rows and columns, resulting in a total of 16 unique treatment combinations. Each treatment combination was replicated across three test tubes, culminating in a total of 48 test tubes as integral components of the experiment.

For the culture experiment, calli of sandalwood, after 30 days, were dissected into small pieces and placed on sterile filter paper. Subsequently, these pieces were cultured on MS media, and three different concentrations of SeNPs as elicitors along with LED elicitation as described above. The experiment utilized 48 autoclaved test tubes (Table 1), each containing approximately 9 mL of medium, and was sealed with a cotton plug following Sardar et al. (2023). In the first column (T1 to T4), there were 12 test tubes, with four treatments and three replicates for each treatment. The treatments in this column included no elicitation (T1) and elicitation with 30 µg/L, 60 µg/L, and 90 µg/L of SeNPs alone (designated as T2, T3, and T4, respectively). The LED elicitation conditions in this column were Control (T1) with white fluorescent light at 15 °C ± 1 °C with an intensity of 3,000 µmol m−2 s−1. Moving to the second column (T5 to T8), again consisting of 12 test tubes, there were four treatments, each with three replicates. In this column, the test tubes were exposed to 100% green LED at ∼550 nm wavelength with a bandwidth of 20 nm at 12 peak height along with one of the three concentrations of SeNPs (30 µg/L, 60 µg/L, and 90 µg/L), labelled as T5, T6, T7, and T8, respectively. In the third column (T9 to T12), another 12 test tubes were arranged with four treatments and three replicates each. These test tubes were subjected to 100% red LED at ∼660 nm wavelength with a bandwidth of 20 nm at 12 peak height along with 30 µg/L, 60 µg/L, and 90 µg/L of SeNPs, forming the treatments T9, T10, T11, and T12, respectively. Lastly, the fourth column (T13 to T16) also comprised 12 test tubes with four treatments and three replicates per treatment. These test tubes were exposed to 100% blue LED at ∼460 nm wavelength with a bandwidth of 20 nm at 12 peak height along with 30 µg/L, 60 µg/L, and 90 µg/L of SeNPs, designated as T13, T14, T15, and T16, respectively. In the LED light elicitation experiments for sandalwood callus cultures, a light intensity of 3,000 µmol m−2 s−1 was maintained, accompanied by a constant temperature of 15 °C ± 1 °C. The photoperiod was set to 16 h of continuous light followed by 8 h of darkness, and the cycling pattern employed was continuous throughout the experimental duration. Each column represented different combinations of SeNPs and LED light, with specific concentrations, creating a comprehensive experimental design with a total of 48 test tubes, each with unique elicitation conditions for further analysis and comparison. After 50 days, data on fresh weight (FW), dry weight (DW), callus moisture content, and number of shoot branches per callus were collected. To ascertain the fresh weight (FW), calli obtained from each treatment underwent cleansing with sterile distilled water. Subsequently, they were positioned on filter paper, compressed with forceps to eliminate surplus water, and weighed. Following this, each callus underwent a 24-hour drying process in an oven set at 50 °C before being re-weighed. The obtained fresh weight (FW) and dry weight (DW) data were converted to grams per litter. The percentage moisture content was determined by employing the FW and DW of the callus, utilizing the formula outlined by Rashmi & Trivedi (2014).

Percentage moisture contents = [(B-A) - (C-A)]/(B-A) X 100.

Letting A represent the weight of the empty petri dish, B denotes the weight of the petri dish with fresh callus, and C signifies the weight of the petri dish with dried callus, observations were meticulously recorded. These observations encompassed the characteristics of callus nature, response and colour. This systematic approach aimed to comprehend the effects of elicitation on the morphological health of the callus.

Extract preparation

Callus cultured in vitro (100 g) was harvested, rinsed with distilled water, freeze-dried, and finely powdered using a mortar and pestle. The resulting powder was then subjected to an 18-hour extraction in dichloromethane: methanol (1:1, v/v) at 40 °C, maintaining a ratio of 1:200 (w/v) of plant material to solvent. After extraction, solid residues were removed through filtration using Whatman No. 1 filter paper and subsequent centrifugation at 5,000 g for 10 min. The resulting supernatant was concentrated utilizing an Eyela N-N series rotary evaporator coupled with an Eyela aspirator (Model: A 3S; Rikakikai Inc., Tokyo, Japan) at 40 °C under reduced pressure. The obtained sandalwood extract was stored at −20 °C for future use. After 50 days of callus cultivation, the extract was prepared for the study of phytochemicals (Misra & Dey, 2012).

Determination of casein/BSA/PVPP-bound tannins

For casein/BSA/PVPP-bound tannins, 100 mg of BSA/casein or PVPP was added to 200 µl of the extract, followed by shaking for 1 h at room temperature and incubation at 4 °C for 1–2 h. The mixture was then filtered or centrifuged, and the filtrate represented unbound polyphenols. Bound tannins were calculated as the difference between total polyphenols and unbound polyphenols, and the results were expressed as milligrams of catechin per gram of the extract (Schneider, 1976)

Determination of total flavan-3-ols content

In the determination of total flavan-3-ols content, a sample solution containing 1 mg/ml of the extract in methanol was prepared. To 200 µl of this sample, 1 ml of p-dimethylamino cinnamaldehyde (DMACA) reagent (0.1% in 1 N HCl in methanol) was added. The mixture was vortexed and allowed to stand at room temperature for 10 min. During this time, a blue color developed, and the intensity was measured at 640 nm using a spectrophotometer. Catechin was used as the standard, ranging from 25 to 150 µg, and the results were expressed as milligrams of catechin per gram of the extract. This method was adapted from the reference provided (Arnous, Makris & Kefalas, 2001).

Determination of total saponin content

To determine the total saponin content, a sample solution containing 1 mg/ml of the extract in methanol was prepared. Ten microliters of this extract were mixed with 50 µl of 8% vanillin in ethanol and 500 µl of 72% H2SO4. The mixture was heated to 60 °C for 20 min and then cooled to 4 °C for 5 min. During this process, a yellow/green color developed, and the absorbance was measured at 544 nm using a spectrophotometer. Sapogenin was used as the standard, ranging from 10 to 100 µg, and the results were expressed as milligrams of sapogenin per gram of the extract. This method was adapted from the reference provided (Makkar, Siddhuraju & Becker, 2007).

Total terpenoid contents

In the total terpenoid content assay, 100 µg of the plant extract dissolved in methanol was used. To this, 500 µl of 2% vanillin- H2SO4 solution in cold methanol was added. The mixture was heated to 60 °C and kept at this temperature for 20 min, then cooled to 25 °C for 5 min. Within the next 20 min, the absorbance of the solution was measured using a spectrophotometer at a wavelength of 608 nm. The color of the solution changed to blue–green, and the absorbance at 608 nm was recorded. Linalool was used as the standard, ranging from 20 to 100 mg/l, and the results were expressed as micrograms of linalool per milligram of the extract. This method was adapted from the reference provided (Misra & Dey, 2012).

Determination of total flavonoid content and total phenolic contents

Sandalwood plant extract’s total flavonoid content (TFC) was evaluated using a modified method by Ebrahimzadeh, Pourmorad & Bekhradnia (2008). The plant extract in methanol was combined with AlCl2 and sodium acetate, and after incubation, absorbance was measured at 415 nm. Quercetin served as a standard for calibration, and the flavonoid content was expressed as milligrams of quercetin equivalent per gram of extract (mg QE g−1). For total phenolic content (TPC) in sandalwood callus extracts, a modified method based on Kim, Jeong & Lee (2003) was employed. The plant extract was mixed with Folin–Ciocalteu reagent and Na2CO3, and after incubation, absorbance was measured at 750 nm. Gallic acid was used as a standard for calibration, and TPC was expressed as milligrams of gallic acid equivalent per gram of extract (mg GA g−1).

Determination of DPPH scavenging percentage

The DPPH free radical scavenging assay was employed to assess the antioxidant activity of sandalwood culture extracts. A combination of plant extract (50 µL) and 3 mL of methanolic DPPH solution (0.004%) was incubated in darkness at room temperature for 30 min. Subsequently, the absorbance was measured at 517 nm using a UV–visible spectrophotometer. The percentage of free radical scavenging activity (%RSA) was then calculated using the following formula:

%RSA = {(Absorbance of control - Absorbance of Sample)/Absorbance of control}×100, with BHT (Sigma-Aldrich) as the standard (Sardar et al., 2023).

Determination of activities of bioactive antioxidants

To evaluate the antioxidant enzyme activity of specific callus samples, a modified version of the methods described by Khan et al. (2013), Giannopolitis & Ries (1977), Arrigoni et al. (1992), and Abeles & Biles (1991) was employed. Each sample was homogenized, and the resulting supernatants were utilized to measure the activities of antioxidant enzymes, including superoxide dismutase (SOD), catalase (CAT), peroxidase (POD), and phenylalanine ammonia-lyase (PAL). PAL activity was assessed through the absorbance fluctuation at 290 nm.

Determination of anthocyanin and tocopherol contents

The callus was dried and pigments were extracted. Chromatography was performed using butanol-acetic acid-water (BAW) in a 4:1:5 ratios, butanol-hydrochloric acid (Bu-HCl) in a 1:1 ratio, and 1% hydrochloric acid (HCl) as described by Yashin & Yashin (2006). Anthocyanin content was measured at a wavelength of 525 nm.

The alpha-tocopherol content was determined by first extracting the compound from the sample using hexane as the solvent. The sample was weighed and homogenized with hexane, and the resulting mixture was filtered to separate the liquid phase containing alpha-tocopherol. The hexane was then evaporated using a rotary evaporator to concentrate the extract. Chromatographic separation was performed with a C18 reversed-phase column, employing a mobile phase consisting of hexane and isopropanol as specified by Baker Jr (1981). Alpha-tocopherol was detected using a UV-Vis spectrophotometer set to 295 nm. A calibration curve was prepared with standards of known alpha-tocopherol concentration, and this curve was used to quantify the alpha-tocopherol content in the sample.

Experimental design and statistical analysis

A randomized experimental design was adopted for all investigations, ensuring a comprehensive and unbiased approach. Each treatment underwent replication three times to enhance the reliability of the results. The collected data were subjected to rigorous analysis through a two-way analysis of variance (ANOVA), utilizing Costat software version 6.3. To determine mean values and assess statistically significant differences (p < 0.05), the Turkey Honestly Significant Difference (HSD) test was applied (Mazhar, Akram & Shahid, 2022; Mazhar et al., 2023). The Turkey HSD test is widely used for post hoc analysis in situations where multiple pairwise comparisons are needed, providing a robust method to identify differences between treatment means while controlling the experiment-wise error rate.

Results

Influence on callus formation and appearance

Under white fluorescent light (WFL) exposure as a control treatment, the callus exhibited a vulnerable nature, with a somewhat lackluster response and a dull yellow coloration. Conversely, the green LED Light (G) treatments showcased a range of callus natures, transitioning from fragile to partially firm and moderately dense. The callus response improved significantly, spanning from poor to very good, and the colors evolved from a golden hue with a touch of brown to yellow-greenish and moderately greenish shades. In the context of red LED Light (R) exposure, the callus displayed a tender nature in lower concentrations of SeNPs (30 µg/L) and a resilient nature in higher concentrations of SeNPs. The callus response exhibited a notable enhancement, progressing from poor to very good, with color variations from yellow-brown to yellow-greenish and finally to bright greenish hues. The callus response, initially poor, showed remarkable improvement, reaching an excellent rating. The color spectrum also shifted from yellow-brown to bright greenish and brilliantly greenish shades upon elicitation with SeNPs and blue light. Specifically, the higher doses of SeNPs (60 and 90 µg/L) were better in callus formation. Similarly, red and blue LED lights were more beneficial in improving callus appearance and formation (Table 2).

Table 2 Callus formation and appearance of sandalwood as elicited and modulated by different concentrations of SeNPs and LED light exposure.

Treatments	Callus nature	Callus
responsb	Callus colour	
0 µg/L + WFLa	Vulnerable	+	Dull yellow	
30 µg/L + WFL	Tender	+	Golden hue with a subtle touch of brown	
60 µg/L + WFL	Fragile	+	Golden hue with a subtle touch of brown	
90 µg/L + WFL	Fragile	+	Golden hue with a subtle touch of brown	
0 µg/L + G	Fragile	++	Golden hue with a subtle touch of brown	
30 µg/L + G	Partially firm	+++	Yellow greenish	
60 µg/L + G	Moderately dense	++++	Moderately greenish	
90 µg/L + G	Moderately dense	++++	Moderately greenish	
0 µg/L + R	Tender	+	Yellow-brown	
30 µg/L + R	Semi Compact	+++	Yellow greenish	
60 µg/L + R	Compact	++++	Greenish	
90 µg/L + R	Resilient	++++	Bright greenish	
0 µg/L + B	Fragile	+	Yellow-brown	
30 µg/L + B	Compact	++++	Bright greenish	
60 µg/L + B	Compact	++++	Very bright greenish	
90 µg/L + B	Resilient	++++	Brilliantly greenish	
Notes.

a WFL, White fluorescent light; G, Green LED light; R, Red LED light; B, Blue LED light.

b ++++, excellent; +++, very good; ++, average; +, poor.

The presence and concentration of SeNPs in the experimental treatments played a significant role in shaping the callus formation and appearance in sandalwood. As observed in the results, treatments with higher concentrations of SeNPs generally led to more favorable callus responses. For instance, under green LED light (G) exposure, callus treated with 90 µg/L of SeNPs exhibited a moderately dense nature and an excellent callus response, indicating that the presence of SeNPs promoted the formation of healthier and more robust callus tissue. Similar trends were observed in red LED light (R) and blue LED light (B) treatments, where higher SeNP concentrations resulted in resilient callus with significantly improved responses, ranging from very good to excellent (Table 2). A pictorial description of the callus has been presented in Fig. 1.

Figure 1 Pictorial description of callus formation in tissue culture of sandalwood.

(A) Callus initiation. (B) Proliferation. (C) Multiplication. (D) Differentiation.

Influence on callus growth and moisture contents

Table 3 provides detailed information on callus growth and moisture content in sandalwood as influenced by different concentrations of SeNPs and LED light exposure. The data is presented in terms of callus fresh weight (in grams), callus dry weight (in grams), callus moisture content (in percentage), and the number of shoots per callus. The results show that as the concentration of SeNPs increases, there is a general trend of increased callus fresh weight across all types of LED light exposure. For example, under blue LED light (B) exposure, the callus fresh weight ranges from 110.77 g (for 0 µg/L + B) to 149.33 g (for 90 µg/L + B), demonstrating a significant increase with higher SeNPs concentration. Similar to callus fresh weight, the data indicates an increase in callus dry weight with higher concentrations of SeNPs. For instance, under red LED light (R) exposure, the callus dry weight varies from 23.87 g (for 0 µg/L + R) to 35.50 g (for 90 µg/L + R), showing a substantial rise with increasing SeNPs concentration. All the treatments improved callus growth and moisture content (Table 3). However, 90 µg/L + B was the most effective. The number of shoots per callus indicates the number of shoots that have developed from each callus tissue. Higher SeNP concentrations, especially under red LED light (R) and blue LED light (B) exposure, result in an increased number of shoots per callus. For example, under red LED light, the number of shoots per callus ranges from 18.00 (for 0 µg/L + R) to 21.46 (for 90 µg/L + R), demonstrating a positive effect of SeNPs on shoot development.

Table 3 Callus growth and moisture contents as elicited and modulated by different concentrations of SeNPs and LED light exposure.

Elicitor concentration	Callus fresh
weight (g)		Callus dry
weight (g)		Callus
moisture
(%)		No. of
shoot/
callus		
	Mean	±SD	Mean	±SD	Mean	±SD	Mean	±SD	
0 µg/L + WFLa	102.67 m	1.53	22.67 k	0.76	48.10 m	0.79	14.53 l	0.21	
30 µg/L + WFL	105.63 kl	0.78	25.20 j	0.72	51.70 k	0.70	16.50 k	0.36	
60 µg/L + WFL	108.67 j	0.76	27.47 i	0.42	55.47 j	0.50	17.30 j	0.26	
90 µg/L + WFL	110.30 j	0.98	28.23 hi	0.25	58.50 i	0.50	18.00 i	0.20	
0 µg/L + G	103.67 lm	1.53	22.67 k	1.45	48.50 lm	0.87	16.90 jk	0.36	
30 µg/L + G	114.53 i	0.96	29.13 h	0.51	61.00 h	0.60	18.90 h	0.20	
60 µg/L + G	119.60 h	1.80	30.73 g	0.50	64.83 g	1.04	19.70 g	0.30	
90 µg/L + G	123.93 g	1.29	31.80 fg	0.20	66.00 g	1.00	20.27 f	0.15	
0 µg/L + R	106.30 k	0.82	23.87 jk	0.81	49.70 l	0.26	18.00 i	0.70	
30 µg/L + R	126.60 f	0.92	32.33 ef	0.31	69.47 f	0.61	20.86 e	0.10	
60 µg/L + R	132.87 e	1.40	33.70 e	0.79	71.53 e	0.61	21.35 d	0.02	
90 µg/L + R	138.10 d	1.15	35.50 d	1.32	74.53 d	0.50	21.46 d	0.09	
0 µg/L + B	110.77 j	1.48	24.90 j	0.17	51.17 k	1.04	19.17 h	0.29	
30 µg/L + B	143.08 c	1.89	41.67 c	1.15	77.93 c	1.01	22.20 c	0.12	
60 µg/L + B	145.23 b	1.37	43.87 b	1.21	79.77 b	0.25	23.77 b	0.21	
90 µg/L + B	149.33 a	0.65	48.33 a	1.15	83.47 a	1.50	24.76 a	0.13	
Notes.

a WFL, White fluorescent light; G, Green LED light; R, Red LED light; B, Blue LED light. Means following different alphabet differ significantly at Turkey’s HSD 5%

Influence of LED and SeNPs elicitation on bioactive antioxidants

Table 4 presents data on the influence of LED elicitation and SeNP treatments on the activities of bioactive antioxidants in the callus extracts of sandalwood. The measured parameters include PAL (phenylalanine ammonia-lyase) activity, superoxide dismutase activity, peroxidase activity, and catalase activity, all expressed in units per gram fresh weight (U/g FW). The data shows that PAL activity increased with higher concentrations of SeNPs and varied with different LED light exposures. For instance, under blue LED light (B) exposure, PAL activity increased from 2.99 U/g FW (for 0 µg/L + B) to 6.46 U/g FW (for 90 µg/L + B), indicating a significant enhancement in the phenylpropanoid pathway due to SeNPs treatment. The results demonstrate an increase in SOD activity with higher SeNP concentrations, especially under green LED light (G) and blue LED light (B) exposures. This suggests that SeNP treatment enhances the plant’s ability to counteract oxidative stress by increasing SOD activity. The data shows that peroxidase activity increased significantly with SeNP treatment, particularly under red LED light (R) and blue LED light (B) exposures. Higher SeNP concentrations led to elevated peroxidase activity, indicating an enhanced capacity for oxidative stress management in the treated callus tissues. The results reveal an increase in catalase activity with higher SeNP concentrations, especially under green LED light (G) and blue LED light (B) exposures. This implies that SeNP treatment enhances the plant’s ability to mitigate oxidative damage by increasing catalase activity.

Table 4 Influence of LED elicitation and SeNPs treatments on the functions of bioactive antioxidants in the callus extracts of sandalwood.

Elicitor concentration	PAL
(U/g FW)		Superoxide
dismutase
(U/g FW)		Peroxidase
(U/g FW)		Catalase
(U/g FW)		
	Mean	±SD	Mean	±SD	Mean	±SD	Mean	±SD	
0 µg/L + WFLa	2.76 k	0.02	1.21 j	0.01	1.54 l	0.01	0.51 j	0.00	
30 µg/L + WFL	2.92 ij	0.02	1.22 j	0.00	1.56 kl	0.00	0.52 ij	0.00	
60 µg/L + WFL	3.01 i	0.07	1.24 j	0.00	1.57 jkl	0.00	0.54 ij	0.01	
90 µg/L + WFL	3.11 h	0.02	1.29 ij	0.01	1.59 jkl	0.00	0.57 hij	0.00	
0 µg/L + G	3.01 i	0.03	1.34 i	0.01	1.62 ijk	0.00	0.61 hi	0.00	
30 µg/L + G	3.15 h	0.09	1.55 h	0.09	1.66 i	0.01	0.64 h	0.01	
60 µg/L + G	4.03 g	0.09	2.01 g	0.08	2.67 g	0.04	0.98 g	0.02	
90 µg/L + G	4.29 f	0.07	3.79 d	0.06	2.82 f	0.06	2.30 e	0.09	
0 µg/L + R	2.83 jk	0.03	1.27 ij	0.02	1.58 jkl	0.00	0.55 ij	0.00	
30 µg/L + R	4.37 f	0.11	2.26 f	0.07	2.16 h	0.07	1.12 f	0.02	
60 µg/L + R	4.86 e	0.05	4.58 b	0.06	3.82 d	0.05	2.48 d	0.17	
90 µg/L + R	5.07 d	0.08	2.66 e	0.06	3.39 e	0.08	2.30 e	0.03	
0 µg/L + B	2.99 i	0.05	1.29 ij	0.00	1.63 ij	0.01	0.57 hij	0.01	
30 µg/L + B	6.09 c	0.05	4.48 c	0.06	4.07 c	0.08	2.76 c	0.06	
60 µg/L + B	6.29 b	0.03	4.64 b	0.04	4.27 b	0.03	3.00 b	0.03	
90 µg/L + B	6.46 a	0.02	4.85 a	0.10	4.54 a	0.03	3.15 a	0.02	
Notes.

a WFL, White fluorescent light; G, Green LED light; R, Red LED light; B, Blue LED light.

Means following different alphabet differ significantly at Turkey’s HSD 5%.

Impact of elicitation treatments on total phenolics, flavonoids, saponins, and anthocyanin accumulation

Table 5 presents the influence of LED elicitation and SeNP treatments on the accumulation of total phenolic, flavonoids, saponins, and anthocyanin in the callus extracts of sandalwood. The data is expressed in milligrams of quercetin equivalents per gram (mg QE/g) for flavonoids, milligrams of gallic acid equivalents per gram (mg GA/g) for phenolics, micrograms of saponin equivalents per milligram (µg saponin E./ mg) for saponins, and milligrams per 100 grams (mg/100 g FW) for anthocyanins. All the elicitation treatments significantly enhanced the mentioned variables. The individual effect of LED and SeNPs was more pronounced when used in synergism. The treatments with the highest flavonoid content include 90 µg/L + B (blue LED light, 90 µg/L SeNPs) with 0.10 mg QE/g, indicating a significant increase in flavonoid accumulation. Phenolics are diverse compounds with various biological activities. The treatments with the highest phenolic content include 90 µg/L + B (blue LED light, 90 µg/L SeNPs) with 19.11 mg GA/g, suggesting a substantial enhancement in phenolic compound accumulation. Similarly, the treatment with the highest saponin content includes 90 µg/L + G (green LED light, 90 µg/L SeNPs) with 6.79 µg saponin E./ mg, indicating a notable increase in saponin accumulation. The treatments with the highest anthocyanin content include 90 µg/L + B (blue LED light, 90 µg/L SeNPs) with 5.96 mg/100 g FW, indicating a significant increase in anthocyanin accumulation. The table shows that the red and blue lights are better at increasing phytochemicals.

Table 5 Influence of LED elicitation and SeNPs treatments on the total phenolics, flavonoids, saponins and anthocyanin accumulation in the callus extracts of sandalwood.

Elicitor concentration	Flavonoids
mg QE/g	Phenolics
mg GA/g	Saponin
µg saponin E./ mg	Anthocyanin
(mg/100 g FW)	
	Mean	±SD	Mean	±SD	Mean	±SD	Mean	±SD	
0 µg/L + WFLa	0.03 j	0.001	10.53 m	0.17	3.81 o	0.04	3.12 m	0.04	
30 µg/L + WFL	0.03 j	0.001	12.42 j	0.10	4.13 n	0.03	3.22 l	0.03	
60 µg/L + WFL	0.04 i	0.001	13.01 i	0.10	4.34 m	0.11	3.43 k	0.03	
90 µg/L + WFL	0.04 i	0.002	13.19 i	0.04	4.84 l	0.07	3.53 j	0.03	
0 µg/L + G	0.03 j	0.001	11.39 l	0.04	4.05 n	0.06	3.25 l	0.05	
30 µg/L + G	0.05 f	0.001	13.69 h	0.14	5.48 j	0.17	3.70 i	0.04	
60 µg/L + G	0.05 f	0.002	13.88 gh	0.09	6.15 h	0.07	3.84 h	0.05	
90 µg/L + G	0.06 e	0.001	14.07 g	0.07	6.79 g	0.09	4.01 g	0.07	
0 µg/L + R	0.04 i	0.001	11.98 k	0.03	5.10 k	0.08	3.47 jk	0.07	
30 µg/L + R	0.07 d	0.002	14.41 f	0.17	7.15 f	0.03	4.39 f	0.07	
60 µg/L + R	0.07 d	0.003	14.72 e	0.05	7.51 e	0.08	4.65 e	0.11	
90 µg/L + R	0.08 c	0.003	15.08 d	0.13	7.96 d	0.05	4.86 d	0.08	
0 µg/L + B	0.04 i	0.003	12.51 j	0.06	5.61i	0.06	3.54 j	0.03	
30 µg/L + B	0.09 b	0.005	16.74 c	0.28	8.10 c	0.11	5.59 c	0.04	
60 µg/L + B	0.09 b	0.006	18.41 b	0.17	8.98 b	0.06	5.82 b	0.05	
90 µg/L + B	0.10 a	0.005	19.11a	0.11	9.19 a	0.03	5.96 a	0.05	
Notes.

a WFL: White fluorescent light; G: Green LED light; R: Red LED light; B: Blue LED light. Means following different alphabet differ significantly at Turkey’s HSD 5%

Influence of elicitation on casein/BSA/PVPP-bound tannins (TCTC), flavan-3-oils, terpenoids, and tocopherol contents

Table 6 presents the influence of LED elicitation and selenium nanoparticles (SeNPs) treatments on the content of casein/BSA/PVPP-bound tannins (TCTC), flavan-3-ols, terpenoids, and tocopherols in the callus extracts of sandalwood. All the treatments significantly enhanced the contents of TCTC, flavan-3-ols, terpenoids, and tocopherols in the callus extracts of sandalwood. However, higher doses of SeNPs and blue and red LED light elicitations were more effective, specifically when they were employed in synergism. Treatments with the highest terpenoid content include 90 µg/L + B (blue LED light, 90 µg/L SeNPs) with 14.88 µg linalool E./mg, indicating a significant increase in terpenoid accumulation. The treatments with the highest flavan-3-ol content include 90 µg/L + B (blue LED light, 90 µg/L SeNPs) with 7.54 µg catechin E./mg, suggesting a substantial enhancement in flavan-3-ol accumulation. The treatments with the highest tannin content include 90 µg/L + B (blue LED light, 90 µg/L SeNPs) with 3.45 µg catechin E./mg, indicating a notable increase in tannin accumulation. Treatments with the highest α-tocopherol content include 90 µg/L + B (blue LED light, 90 µg/L SeNPs) with 4.90 µg/g FW, demonstrating a significant enhancement in tocopherol accumulation. The treatments with 90 µg/L of SeNPs combined with blue LED light (90 µg/L + B) appear to be the most effective in promoting the accumulation of terpenoids, flavan-3-ols, tannins, and tocopherols in sandalwood callus extracts. These treatments demonstrate significant enhancements in the levels of these bioactive compounds, indicating their potential applications in the pharmaceutical, cosmetic, and food industries due to their medicinal and antioxidant properties.

Table 6 Influence of LED elicitation and SeNPs treatments on casein/BSA/PVPP-bound tannins (TCTC), flavan-3-ols, terpenoids, and tocopherol contents in the callus extracts of sandalwood.

Elicitor concentration	Terpenoids
µg linalool E./mg	Flavan-3-ols
µg catechin E./mg	Tannins
µg catechin E./mg	Tocopherols
(µg/g FW)	
	Mean	±SD	Mean	±SD	Mean	±SD	Mean	±SD	
0 µg/L + WFLa	6.30 m	0.17	2.16 o	0.04	0.61 m	0.04	3.38 m	0.06	
30 µg/L + WFL	8.19 j	0.10	2.48 n	0.03	0.71 l	0.03	3.57 l	0.05	
60 µg/L + WFL	8.78 i	0.10	2.69 m	0.11	0.92 k	0.03	3.66 k	0.03	
90 µg/L + WFL	8.96 i	0.04	3.19 l	0.07	1.02 j	0.03	3.69 j	0.02	
0 µg/L + G	7.16 l	0.04	2.40 n	0.06	0.74 l	0.05	3.78 l	0.02	
30 µg/L + G	9.46 h	0.14	3.83 j	0.17	1.19 i	0.04	3.85 i	0.04	
60 µg/L + G	9.65 gh	0.09	4.50 h	0.07	1.33 h	0.05	3.95 h	0.02	
90 µg/L + G	9.84 g	0.07	5.14 g	0.09	1.50 g	0.07	4.10 g	0.05	
0 µg/L + R	7.75 k	0.03	3.45 k	0.08	0.96 jk	0.07	3.75 jk	0.04	
30 µg/L + R	10.18 f	0.17	5.50 f	0.03	1.88 f	0.07	4.14 f	0.07	
60 µg/L + R	10.49 e	0.05	5.86 e	0.08	2.14 e	0.11	4.27 e	0.03	
90 µg/L + R	10.85 d	0.13	6.31 d	0.05	2.35 d	0.08	4.39 d	0.03	
0 µg/L + B	8.28 j	0.06	3.96 i	0.06	1.03 i	0.03	3.84 i	0.03	
30 µg/L + B	12.51 c	0.28	6.45 c	0.11	3.08 c	0.04	4.64 c	0.06	
60 µg/L + B	14.18 b	0.17	7.33 b	0.06	3.31 b	0.05	4.72 b	0.03	
90 µg/L + B	14.88 a	0.11	7.54 a	0.03	3.45 a	0.05	4.90 a	0.07	
Notes.

a WFL, White fluorescent light; G, Green LED light; R, Red LED light; B, Blue LED light.

Means following different alphabet differ significantly at Turkey’s HSD 5%.

Figure 2 shows the DPPH scavenging percentage of the callus extract of sandalwood elicited with LED light and various elicitation doses of SeNPs. The DPPH scavenging percentage is a measure of the ability of the extract to scavenge DPPH, a free radical. A higher DPPH scavenging percentage indicates that the extract is more effective at scavenging free radicals. The DPPH scavenging percentage of the extract increased with increasing elicitation doses of SeNPs. This suggests that SeNPs can enhance the antioxidant activity of sandalwood extracts. The figure also shows that the DPPH scavenging percentage of the extract was higher when the callus cultures were elicited with LED light than when they were not elicited with LED light. This suggests that LED light can also enhance the antioxidant activity of sandalwood extracts. Treatment 16 involving the combined use of 90 µg/L SeNPs and blue light was most significant in increasing the DPPH scavenging capacity of the callus extract.

Figure 2 DPPH Scavenging percentage of the callus extract of sandalwood elicited with LED light and various elicitation doses of SeNPs.

On the x-axis, T1 to T16 indicate the treatment numbers as mentioned in Table 1. Bars representing means that have different letters are significantly different from each other.

Statistical explanation of correlation matrix, ANOVA, and principal component analysis

The results from the two-way ANOVA (Table 7) indicate that both SeNP concentrations and the type of LED light exposure have significant effects on various biochemical and physiological parameters in sandalwood callus cultures. The interaction between SeNPs and LED light further contributes to the observed variations, highlighting the complex interplay between these factors in shaping the biochemical composition and antioxidant potential of sandalwood callus extracts. Kendell’s correlation matrix is shown in Table 8. There are strong positive correlations (ranging from 0.90 to 1) between various biochemical parameters indicating that these variables tend to increase or decrease together in response to the treatments. A strong positive correlation (0.94 to 0.99) was observed for growth variables, suggesting that these growth-related parameters are closely related and respond similarly to the experimental conditions. There are no significant negative correlations among the studied variables, indicating that none of the variables move in the opposite direction under the experimental conditions.

Table 7 Two-way analysis of variance (ANOVA) results of the variables studied in the experiment as affected by LED elicitation and SeNPs treatments.

Variation source	a df	CFW	CDW	CMC	NSB/C	TFC	TOCO	
Row-wise treatment effect (F1)	3	1404.64b***(0.000)	359.18***(0.000)	1087.63***(0.000)	36.705***(0.000)	0.003***(0.000)	0.757***(0.000)	
Column-wise treatment effect (F2)	3	2062.71***(0.000)	428.11***(0.000)	849.10***(0.000)	73.812***(0.000)	0.004***(0.000)	1.896***(0.000)	
Interaction (F 1 ×F2)	9	128.46*** (0.000)	34.818***(0.000)	67.763*** (0.000)	0.804*** (0.000)	2. 3 ×10−4***(0.000)	0.084***(0.000)	
Error	32	1.591	0.704	0.643	0.077	0.00001	0.001	
Variation Source	df	TPC	TSC	TTC	FLAVO	TCTC	ANTHO	
Row-wise treatment effect (F1)	3	34.780***(0.000)	14.824***(0.000)	34.780*(0.000)	14.824***(0.000)	3.699***(0.000)	3.699***(0.000)	
Column-wise treatment effect (F2)	3	42.906*** (0.000)	30.758***(0.000)	42.906***(0.000)	30.759***(0.000)	8.319*** (0.000)
	8.319***(0.000)	
Interaction (F 1 ×F2)	9	2.201 *** (0.000)	0.912***(0.000)
	2.201 *** (0.000)	0.912*** (0.000)	0.565*** (0.000)	0.565***(0.000)	
Error	32	0.015	0.006	0.015	0.006	0.003	0.003	
Variation Source	df	PAL	SOD	POD	CAT	DPPH	
Row-wise treatment effect (F1)	3	8.190***(0.000)	9.230***(0.000)	6.076*** (0.000)	5.254*** (0.000)	1849.0***(0.000)	
Column-wise treatment effect (F2)	3	13.719*** (0.000)	13.822***(0.000)	9.175***(0.000)	7.209***(0.000)	3156.72***(0.000)
	
Interaction (F 1 ×F2)	9	1.443*** (0.000)	2.948*** (0.000)	1.282*** (0.000)	1.211*** (0.000)	71.481*** (0.000)	
Error	32	0.003	0.002	0.001	0.001	3.3111	
Notes.

a df. Degree of freedom.

*** Significant at 0.001 level.

PAL Phenylalanine ammonia-lyase

SOD Superoxide dismutase

Toco Tocopherols

Antho Anthocyanin

TFC Total flavonoid contents

CAT Catalase

CMC Callus moisture contents

TPC Total phenolics contents

POD Peroxidase

DPPH 2,2-diphenyl-1-picrylhydrazyl

CFW Callus fresh weight

CDW Callus dry weight

NSB/C Number of shoot branches per callus

TTC Terpenoids contents

TSC Total saponin contents

TCTC casein/BSA/PVPP-bound tannins

Flavo flavan-3-ols

Table 8 Kendell’s correlation matrix of the studied variables from the callus extracts of sandalwood treated with LED light elicitation and SeNPs.

Variables	CFW	CDW	CMC	NSB/C	TFC	TPC	TSC	TTC	Flavo	TCTC	PAL	SOD	POD	CAT	DPPH	Toco	Antho	
CFW	1																	
CDW	0.98*																	
CMC	0.98*	0.99*																
NSB/C	0.98*	0.94*	0.95*															
TFC	0.97*	0.98*	0.98*	0.98*														
TPC	0.99*	0.99*	0.98*	0.96*	0.99*													
TSC	0.98*	0.95*	0.95*	0.99*	0.98*	0.96*												
TTC	0.99*	0.99*	0.98*	0.96*	0.99*	0.98*	0.96*											
Flavo	0.98*	0.95*	0.95*	0.99*	0.98*	0.96*	0.98*	0.96*										
TCTC	0.99*	0.97*	0.97*	0.99*	0.99*	0.97*	0.99*	0.97*	0.99*									
PAL	0.97*	0.97*	0.97*	0.95*	0.97*	0.98*	0.94*	0.98*	0.94*	0.96*								
SOD	0.92*	0.90*	0.90*	0.94*	0.93*	0.91*	0.93*	0.91*	0.93*	0.94*	0.94*							
POD	0.95*	0.92*	0.92*	0.96*	0.95*	0.93*	0.95*	0.93*	0.95*	0.96*	0.95*	0.98*						
CAT	0.93*	0.92*	0.92*	0.95*	0.94*	0.93*	0.93*	0.93*	0.93*	0.95*	0.96*	0.98*	0.99*					
DPPH	0.98*	0.96*	0.96*	0.99*	0.99*	0.97*	0.99*	0.97*	0.99*	0.99*	0.95*	0.93*	0.95*	0.94*				
Toco	0.95*	0.91*	0.92*	0.98*	0.95*	0.93*	0.97*	0.93*	0.97*	0.96*	0.94*	0.96*	0.98*	0.97*	0.97*			
Antho	0.99*	0.97*	0.97*	0.99*	0.99*	0.97*	0.99*	0.97*	0.99*	0.97*	0.96*	0.94*	0.96*	0.95*	0.99*	0.96*	1	
Notes.

* Significantly differ from zero at alpha 0.05. PAL: Phenylalanine ammonia-lyase

SOD Superoxide dismutase

Toco Tocopherols

Antho Anthocyanin

TFC Total flavonoid contents

CAT Catalase

CMC Callus moisture contents

TPC Total Phenolics contents

POD Peroxidase

DPPH 2,2-diphenyl-1-picrylhydrazyl

CFW Callus fresh weight

CDW Callus dry weight

NSB/C Number of shoot branches per callus

TTC Terpenoids contents

TSC Total saponin contents

TCTC casein/BSA/PVPP-bound tannins

Flavo flavan-3-ols

Figure 3 is a principal component analysis (PCA) biplot, designed to visualize the relationships between variables in a dataset. The biplot displays two principal components (PCs), which are represented by the axes of the biplot, and the variables are represented by vectors. The length of a vector indicates the variable’s contribution to the corresponding PC, and the angle between two vectors indicates their correlation. In this particular biplot, most of the variables are located in the positive quarter of the plot. This indicates that they tend to increase or decrease together. For example, CAT, SOD, and POD all tend to increase together. The two PCs that are displayed in the biplot account for 98.32% of the variance in the data. This means that they can capture most of the important information in the dataset.

Figure 3 The PCA biplot displays variables located in the positive quarter, indicating that an increase in one variable has a corresponding increasing effect on the other variable.

PAL, phenylalanine ammonia-lyase; SOD, superoxide dismutase; Toco, tocopherols; Antho,Anthocyanin; TFC, total flavonoid contents; CAT, catalase; CMC, callus moisture contents; TPC, total phenolics contents; POD, peroxidase; DPPH, 2,2-diphenyl-1-picrylhydrazyl; CFW, callus fresh weight; CDW, callus dry weight; NSB/C, number of shoot branches per callus; TTC, terpenoids contents, TSC, total saponin contents, TCTC, casein/BSA/PVPP-bound tannins; Flavo, flavan-3-ols.

Discussion

The use of nanoparticles and LED light as elicitors to enhance secondary metabolite production, including saponins and terpenoids, in plant cell cultures is an area of active research (Anjum et al., 2021). Elicitors are external factors or substances that induce biochemical or physiological changes in plants, leading to the production of specific secondary metabolites (Patel & Karishnamurthy, 2013). SeNPs have been shown to have a significant impact on callus formation and appearance in the plant tissue culture of sandalwood. Studies have demonstrated that SeNPs can enhance callus induction, promote callus growth, and improve callus quality (Khai et al., 2024; Darwesh, Hassan & Abdellatif, 2023). SeNPs have been found to increase the frequency of callus induction from various plant explants. This effect is attributed to the ability of SeNPs to stimulate cell division and differentiation. SeNPs can also act as a stress protectant, helping to reduce the negative effects of tissue culture on plant cells. SeNPs can also promote the growth of callus cultures (Khai et al., 2022). This is likely due to the enhanced nutrient uptake and metabolic activity that SeNPs can induce in plant cells. SeNPs can also increase the production of plant growth regulators (Gudkov et al., 2020), which further stimulate callus development. SeNPs can improve the overall quality of callus cultures. Better callus coloration with a greenish tinge of sandalwood might be due to the role of SeNPs in the synthesis of pigments. Some studies have reported that SeNPs can induce the production of chlorophyll (Shah et al., 2022), which gives callus a reddish or purple color. SeNPs also increased the compactness of callus cultures of sandalwood in the present study. This is likely due to the enhanced production of cell wall components, such as cellulose and lignin (Wadhwani et al., 2016). A more compact callus is often healthier and more responsive to regeneration treatments. Overall, SeNPs have been shown to have a positive impact on callus formation and appearance. These effects make SeNPs a promising tool for plant tissue culture and biotechnology applications. In the present work, LED exposure particularly the blue and red LEDs led to a significant impact on sandalwood callus color, appearance, and formation. LEDs have emerged as a promising alternative to traditional fluorescent and incandescent lighting in plant tissue culture due to their energy efficiency, long lifespan, and precise control over the light spectrum (Wadhwani et al., 2016). Among the different wavelengths of light, red and blue LEDs have been extensively studied for their effects on callus formation and appearance (Rehman et al., 2020). Red light has been shown to promote callus induction and growth, particularly in the early stages of culture. This is attributed to the role of red light in stimulating cell division and differentiation. Red light can also enhance the production of auxins, which are plant hormones that play a crucial role in callus formation (Zou et al., 2019). The color of the callus can be influenced by the type of light used during culture (Lian et al., 2019; Hassanpour, 2022). LEDs elicitations and SeNPs led to increased callus moisture contents. This might be due to the role of red and blue LEDs in promoting the synthesis of cell wall components, such as cellulose and lignin, which can help to retain moisture in the callus. Some studies have shown that SeNPs can increase the permeability of cell membranes, which could lead to increased callus moisture contents of the callus (Ali et al., 2023).

The functions of bioactive antioxidants SOD, PAL, CAT, and POD were improved by callus elicitation with LED and SeNPs. The effect of blue and red LEDs was more pronounced compared to green LEDs. Blue light, typically in the range of 400-500 nanometers, is known to stimulate photoreceptors like cryptochromes and phototropins in plant cells (Fantini & Facella, 2020). Exposure to blue light can enhance the activity of enzymes involved in various physiological processes, including photosynthesis and defense mechanisms such as SOD. SOD is an antioxidant enzyme that helps scavenge superoxide radicals. It might be assumed that the use of blue LED in the callus cultures of sandalwood might have stimulated photoreceptors leading to improved metabolism. This overall effect led to increased activities of PAL, CAT, POD, and SOD. Red light, typically in the range of 600-700 nanometres, is often used for promoting growth and development in plants. Exposure to red light can influence the expression of genes related to stress responses and defense mechanisms (Guo et al., 2022). POD and CAT are enzymes involved in the detoxification of hydrogen peroxide (H2O2), a type of ROS. Red light exposure may enhance the activities of POD and CAT, thus aiding in H2O2 detoxification (Sharif et al., 2023). Similarly, both blue and red light can influence PAL activity by affecting the expression of genes related to secondary metabolite biosynthesis. The specific response may vary depending on the plant species and the type of callus culture (Endo et al., 2022). The use of SeNPs also increased the functions of these bioactive antioxidants in the callus extracts of sandalwood. The improvement in the antioxidant profile of the callus culture by SeNP can be attributed to their dual role as both a nutrient and a stress elicitor. Selenium, an essential micronutrient provided by SeNPs, is a key component of selenoproteins that have potent antioxidant functions, such as glutathione peroxidases and thioredoxin reductases. These enzymes play a crucial role in neutralizing reactive oxygen species (ROS) and reducing oxidative stress. However, SeNPs also act as abiotic stress elicitors, inducing a mild oxidative stress in the callus culture. This controlled stress leads to a transient increase in ROS levels, which, in turn, act as signalling molecules, triggering the activation of stress response pathways. Consequently, the callus culture upregulates its antioxidant defenses, including the production and activity of antioxidant enzymes. SeNPs could potentially modulate the phenylpropanoid pathway and increase the production of phenolic compounds. PAL activity might be influenced as part of this pathway regulation, although the specific mechanisms would need further investigation (Keerthi Sree et al., 2023).

The use of SeNPs and LEDs as elicitors either individually or in combination significantly impacted the accumulation of various secondary metabolites studied in the callus extracts of sandalwood. LED light, particularly red light, has been associated with increased total phenolic content in plants (Gam et al., 2020). This is attributed to the role of red light in stimulating the phenylpropanoid pathway (Park et al., 2020), the metabolic pathway responsible for phenolic compound biosynthesis. SeNPs, on the other hand, have shown mixed effects on total phenolic accumulation. Some studies have reported increased phenolic content under SeNP treatment, while others have observed no significant effect or even a decrease. The variability in these results might be attributed to plant species, SeNP concentration, and exposure duration (Zhou et al., 2022). Similar to total phenolics, red LED light has been shown to enhance total flavonoid content in callus extracts of sandalwood. This is due to the activation of flavonoid biosynthesis genes under red light exposure (Adil, Abbasi & Haq, 2019). SeNPs have also demonstrated the ability to increase total flavonoid content in plants. This is likely mediated by the antioxidant properties of SeNPs, which can protect flavonoids from degradation and enhance their biosynthesis. The effects of LED light and SeNPs on saponin accumulation are less well-studied compared to phenolic and flavonoids. However, some studies suggest that red LED light can increase saponin content in certain plant species (Watcharatanon, Ingkaninan & Putalun, 2019). Further research is needed to clarify the role of SeNPs in saponin biosynthesis. LED light, particularly blue light, has been shown to induce anthocyanin accumulation in plants. This is due to the activation of anthocyanin biosynthesis genes under blue light exposure (Liu et al., 2022). SeNPs have also demonstrated the ability to increase anthocyanin content in plants. This is likely mediated by the stress-protective effects of SeNPs, which can stimulate anthocyanin production as a response to environmental stressors. Blue light (wavelength around 400–500 nm) is known to regulate several aspects of plant growth and development, including photo-morphogenesis and phototropism. Blue light receptors, such as cryptochromes and phototropins, are involved in signal transduction pathways that influence gene expression. Blue light exposure can upregulate genes encoding enzymes in the phenylpropanoid pathway. For example, it can enhance the expression of genes coding for enzymes like PAL as shown by our results, which is the key enzyme in the phenylpropanoid pathway. PAL catalyzes the conversion of phenylalanine to cinnamic acid, a crucial step in the synthesis of various phenolic compounds (Thoma et al., 2020).

The contents assayed phytoconstituents from the callus extracts of sandalwood such as casein/BSA/PVPP-bound tannins (TCTC), flavan-3-ols, terpenoids, and tocopherol contents increased significantly due to combined and individual elicitation with LEDs and SeNPs. Nanoparticles can enter plant cells and interact with cellular components, influencing various signalling pathways (Heikal et al., 2023). Presumably, these interactions triggered the expression of genes related to secondary metabolite biosynthesis. SeNPs can generate reactive oxygen species (ROS) in plant cells. Moderate levels of ROS can act as signalling molecules, activating defence mechanisms and secondary metabolite production, including saponins and terpenoids. Similarly, LED light of specific wavelengths, such as blue and red, can regulate gene expression and enzymatic activities involved in secondary metabolite biosynthesis pathways. Phytochromes and cryptochromes are photoreceptors that perceive red and blue light, respectively. Activation of these photoreceptors can influence the expression of genes associated with the phenylpropanoid and terpenoid biosynthesis pathways (Weremczuk-Jeżyna et al., 2021). The intensity and duration of light exposure are critical factors. Specific wavelengths of light, such as blue and red, can upregulate the expression of genes encoding enzymes involved in tocopherol biosynthesis. Key enzymes in the tocopherol biosynthesis pathway include homogentisate phytyltransferase (HPT) and tocopherol cyclase (TC). It can be assumed that SeNPs in combination with LEDs increased gene expression leading to higher enzyme activity, promoting the conversion of precursor compounds into tocopherols (Singh et al., 2023). The contents of flavan-3-ols boosted upon the elicitation treatments. In this respect, literature (Hassanvand et al., 2021) reports that nanoparticles can modulate the activity of enzymes involved in flavonoid biosynthesis. Enzymes such as chalcone synthase (CHS), chalcone isomerase (CHI), flavonol synthase (FLS), and flavonoid 3′-hydroxylase (F3′H) are key enzymes in the flavonoid biosynthesis pathway. It might be assumed that LED modulations and SeNP elicitation might have influenced the expression or activity of these enzymes, leading to increased production of flavonols.

While the current study presents a novel approach to enhancing secondary metabolite production in sandalwood callus cultures through the combined application of SeNPs and LED light, it is essential to acknowledge certain limitations and propose avenues for future research. Firstly, the study primarily focused on in vitro callus cultures, and extrapolating these findings to the whole plant may require further investigation. Additionally, future research could delve into elucidating the molecular and biochemical pathways involved in the elicitation process. Investigating the transcriptomic and metabolomics responses of sandalwood under different conditions would provide valuable insights into the intricate mechanisms governing secondary metabolite biosynthesis. Furthermore, exploring the long-term effects of SeNP-LED elicitation on the overall health and viability of sandalwood plants would contribute to a more comprehensive understanding of the approach’s sustainability and practicality for large-scale cultivation. Another avenue for future exploration lies in assessing the potential variations in metabolite profiles among different sandalwood genotypes. Understanding how diverse genetic backgrounds respond to SeNP-LED elicitation could facilitate the development of tailored strategies for optimizing secondary metabolite yields across different sandalwood varieties. Moreover, the study did not explicitly address the economic feasibility and scalability of the proposed elicitation approach. Future research should include a comprehensive economic analysis and consider the practical aspects of implementing the SeNP-LED elicitation strategy on a larger scale, taking into account factors such as cost-effectiveness, resource availability, and environmental sustainability.

From the above discussion, it is clear that SeNPs can act as elicitors, while light serves as a modulator, enhancing the response initiated by the nanoparticles. The innovation behind this work was the strategic integration of SeNPs and LED light, presenting a novel method to amplify metabolite production in plant tissue cultures. The study explored the intricate dynamics between SeNPs and specific LED wavelengths, showcasing a previously unexplored synergy that significantly enhances the growth of callus cultures and the production of key secondary metabolites. The application of higher doses of SeNPs, particularly in conjunction with blue and red LED lights, demonstrated effectiveness in promoting callus growth parameters and elevating the concentrations of bioactive antioxidants and essential secondary metabolites. This innovative elicitation approach not only contributes to overcoming challenges in plant metabolite production but also underscores the potential for advancing sustainable practices in the pharmaceutical and cosmetic industries through improved plant biotechnology methodologies. The effectiveness of these elicitation strategies depends on optimizing the nanoparticle type, concentration, light intensity, and duration of exposure for a particular plant species and callus culture. So far the results were according to our hypothesis.

Conclusion

Our study successfully demonstrated the synergistic enhancement of secondary metabolite production in sandalwood (Santalum album) in vitro callus cultures through the innovative combination of nano selenium (SeNPs) and LED light elicitation. Through experimentation, we established that higher doses of SeNPs, especially when paired with blue and red LED lights, significantly augmented callus growth and the production of valuable secondary metabolites. The application of blue LED light at 90 µg/L SeNPs emerged as the most effective treatment, enhancing various callus growth parameters and substantially increasing the concentrations of bioactive antioxidants. Moreover, this synergistic approach led to a remarkable rise in essential secondary metabolites such as total phenolic, total saponins, casein/BSA/PVPP-bound tannins, flavan-3-ols, and tocopherols. The nanoparticles can mediate an increase in secondary metabolite production in plants by inducing the generation of reactive oxygen species (ROS). These ROS act as signalling molecules that trigger defense pathways and stress responses, ultimately leading to the upregulation of secondary metabolite biosynthesis pathways. These findings highlight the potential of our novel elicitation strategy for significantly improving the efficiency of secondary metabolite production in plant tissue cultures. The implications of this study extend to diverse industries, offering valuable insights for pharmaceuticals, cosmetics, and other sectors reliant on plant-derived compounds. The significance of this study lies in its potential to revolutionize sandalwood cultivation practices, offering a sustainable and efficient approach to enhance secondary metabolite production. The implications of this research extend beyond sandalwood, influencing the broader landscape of plant biotechnology and its applications in various industries.

Supplemental Information

Supplemental Information 1 Supplementary material for PCA chart.

Supplemental Information 2 Raw data.

Additional Information and Declarations

Competing Interests

Author Contributions

Data Availability

The authors declare there are no competing interests.

Muhammad Waqas Mazhar conceived and designed the experiments, performed the experiments, analyzed the data, prepared figures and/or tables, and approved the final draft.

Muhammad Ishtiaq conceived and designed the experiments, analyzed the data, authored or reviewed drafts of the article, and approved the final draft.

Mehwish Maqbool conceived and designed the experiments, analyzed the data, authored or reviewed drafts of the article, and approved the final draft.

Faisal Iqbal Jafri conceived and designed the experiments, analyzed the data, prepared figures and/or tables, and approved the final draft.

Manzer H Siddiqui performed the experiments, analyzed the data, prepared figures and/or tables, and approved the final draft.

Saud Alamri performed the experiments, analyzed the data, prepared figures and/or tables, and approved the final draft.

Mohd Sayeed Akhtar performed the experiments, analyzed the data, prepared figures and/or tables, and approved the final draft.

The following information was supplied regarding data availability:

The raw data are available in Supplementary File.

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
