# Peer review of "Synergistic effects of selenium nanoparticles and LED light on enhancement of secondary metabolites in sandalwood (Santalum album) plants through in-vitro callus culturing technique"

_PeerJ, doi:10.7717/peerj.18106_

## Round 0.1 · original submission · Major Revisions

Both reviewers raise important points. I think that Supplementary material is still important. Besides this, all points raised by both reviewers need to be addressed.

Reviewer 1 ·

Basic reporting

Thanks for your invitation to review the submission by Mazhar, W, M et al (#98799). The authors studied the alteration of metabolic phenotypes of of callus cultures in sandalwood plants by using selenium nanoparticles along with different light regimes. Overall, this research provided new insights into SeNPs and LED light altered metabolism in Sandalwood, which could help expand our knowledge in the field of metabolomics, tissue culture and medicine. Authors has provided logical justification of the study, have used appropriate and reproducible methodology and prominent results with discussion and conclusion. Prior to the consideration of acceptance by the editors, I still have some minor concerns which should be addressed by the authors

Experimental design

1. Author has provided plenty of detail in introduction section, which can be reduced as it should be clear, concise addressing the research gap and hypothesis of the study.
2. Although author has purchased SeNPs but not provided any detail about the characterization of the selenium nanoparticles. Details can be provided present on the supplier websites etc
3. How authors selected the dose of SeNPs and LED light, is there is any previous studies conducted on the same or different plant or, any pilot study conducted before to check the dose response? in most of the studies 10-100 ppm are used?
4. Author should depict the different light types used in the experimentation (pictorial), in the manuscript or as the supplementary material along with the manuscript
5. SeNPs are usually not completely soluble in the water, author should clarify how the solubility was assured, medium used, and how much amount was applied.
6. On the basis of results, it is claimed that SeNPs along with LED light improved secondary metabolites, can author provide some sort of justification that how SeNps can improve metabolic profile?
7. Author should also provide the underlying causes at molecular level that, what might be the mechanism of improving phenolics and flavonoids in the cultures, for example, which pathways are affected which leads to ultimate improvement
8. Selenium usually competes with sulfur, during the incorporation in different amino acids and activation of different enzymes, what might be the possible reason that selenium is more effective compared to sulfur containing amino acids and enzymes?
9. SeNPs improved the antioxidant profile of the callus culture, whether, how this can be explained, that SeNPs also acting as a stress elicitor, which might result in enhanced production of ROS. Can author provide some mechanistic detail about the phenomenon?
10. Figures, especially figure 1, should be improved as it is not visible, it is suggested to provide sketch of the study alone in a table form and these pictures can be given separately
11. Please correct spelling mistakes like, its “Tukeys” nor Turkeys test

Validity of the findings

No Comments

Additional comments

No Comments

Reviewer 2 ·

Basic reporting

'no comment'

Experimental design

'no comment'

Validity of the findings

'no comment'

Additional comments

The present article investigation on aims to optimize the concentration of SeNPs by determining the most effective elicitation dose through a systematic exploration of varying concentrations. Additionally, the study seeks to comprehensively examine the influence of different LED light spectra, including UV, blue, red, and white light, on the biosynthesis of sandalwood oil and related compounds. The investigation of synergistic interactions between SeNPs and specific LED light wavelengths will shed light on whether simultaneous application enhances growth and metabolite yields compared to individual treatments. The study further aims to quantify and characterize key secondary metabolites to provide a detailed understanding of the biochemical changes induced by the combined elicitation approach. Ultimately, by addressing these objectives, the research endeavors to contribute valuable insights to the field of plant biotechnology, offering practical advancements for optimizing sandalwood cultivation practices and enhancing the overall efficiency of secondary metabolite production in vitro cultures. The topic of the manuscript is sound. However, there are some issues that authors must attend to prior to publication.
abstract and Introduction are very long. I think if you make some reduction in introduction, it will be much better.
Supplementary files are not necessary, you already involved in main text. The content is very important for readers and other researchers. It is necessary in scientific structure.
What was the rationale for using different solvents and polarities for the fractionation? Why not evaluate and characterize the compounds that could be potentially isolated?
What were the analytical conditions for spectrometry?
conclusion needs to be strengthened.
Carefully check the space error throughout the manuscript.
All species Latin names should be italicized. Please go through the whole text of the manuscript and do needed changes. The same concerns Latin phrases see : “in vitro”
Reference list should be prepared strictly according to guide for Authors. There are many editorial mistakes in it. There is impossible to mention all of them. There are only some examples: Once each word in the manuscript tile are written with capital letter but the other time not. Please go through very carefully through the whole reference list doing needed changes
Article totally need improve.

Lines 111- 117: please add reference.

---

## Round 0.2 · Minor Revisions

Please, address all comments of reviewer 3.

Reviewer 1 ·

Basic reporting

Basic reporting of the manuscript is up to the mark and according to journal requirements

Experimental design

Experimental design is now improved and well explained

Validity of the findings

Findings are now validated by previous studies and discussion section

Additional comments

Revised manuscript can be accepted

Reviewer 2 ·

Basic reporting

All comments responded by authors. The article is accepted in its present form.

Experimental design

it is approved

Validity of the findings

it is approved

·

Basic reporting

no comment

Experimental design

no comment

Validity of the findings

no comment

Additional comments

Abstract:
• The phrase "The yield and concentration of secondary metabolites (SMs) are variable in plants which is connected with numerous challenges" could be rewritten for better clarity and grammar.
• "novel elicitation approaches have been practiced" could be more specific about the nature of the approaches.
• "dose values of 30 µg/L, 60 µg/L, and 90 µg/L" could be clearer if written as "doses of 30 µg/L, 60 µg/L, and 90 µg/L."
• "a significant synergistic interaction between SeNPs and LED light wavelengths proved an effective protocol" could be rephrased for better flow.
• The phrase "do experience enhanced yield" could be improved for better readability.
• The abstract lacks specific details on the methodology used, such as the controls and statistical analysis.
Introduction:
• The sentence "When used as elicitors, SeNPs have been shown to stimulate the synthesis of secondary metabolites, including antioxidants and phytochemicals, in plants" could be clearer.
Materials and Methods:
• Consider breaking down the steps into bullet points or numbered lists for clarity and ease of understanding.
• Clearly state the rationale behind selecting a temperature of 18 ± 1 °C and the specific light conditions. Explain how these conditions were chosen to support the growth of the Sandalwood plant material.
• The phrase "observed the characteristics of callus nature, response and colour" may benefit from rephrasing for better flow.
• The phrase "The extract was prepared for the study of phytochemicals after 50 days of callus cultivation" might benefit from restructuring for better flow.
• Ensure that the citation for Sardar et al., (2023) is appropriately formatted according to the required citation.
• Can you provide expansions or explanations for the abbreviations BAW, Bu-HCl, and HCI used in the chromatography process for determining anthocyanin content?
• Could you elaborate on the specific steps involved in determining alpha-tocopherol content according to the method outlined by Baker in 1981?
Results and Discussion:
• Can you provide insights into the potential biological mechanisms or pathways that may be responsible for the significant increase in TCTC, flavan-3-ols, terpenoids, and tocopherols observed in sandalwood callus extracts following LED elicitation and SeNPs treatments?
• In the PCA biplot analysis, how were the variables chosen and represented in the plot? Were any data preprocessing steps, such as normalization or scaling, applied to the dataset before conducting the PCA?
• Explore the mechanisms through which SeNPs stimulate cell division and differentiation, enhance nutrient uptake, metabolic activity, and the production of plant growth regulators.

---

## Round 0.3 · accepted · Accept

I confirm that the Authors have addressed all of the reviewers' comments and the manuscript is ready for publication.

·

Basic reporting

no comment

Experimental design

no comment

Validity of the findings

no comment

Additional comments

no comment